# A population estimation study reveals a staggeringly high number of cattle on the streets of urban Raipur in India

**Bhupendra Kumar Sahu**[1], **Arti Parganiha**[1,2]*, **Atanu Kumar Pati**[1,2,3]

**1** School of Studies in Life Science, Pandit Ravishankar Shukla University, Raipur, Chhattisgarh, India,
**2** Center for Translational Chronobiology, Pandit Ravishankar Shukla University, Raipur, Chhattisgarh, India,
**3** Gangadhar Meher University, Sambalpur, Odisha, India

\* arti.parganiha@gmail.com

**Data Availability Statement:** All relevant data are within the manuscript and its Supporting Information files.

**Funding:** The author(s) did not receive any specific funding for this work. However, this work is a part

## Abstract

Cattle are cosmopolitan in distribution. They are economically and ecologically significant. The cattle menace on the urban streets of developing and underdeveloped countries is challenging. The number of road accidents is increasing rapidly over time, in the urban areas of most of the developing countries, like India. In the present study, we estimated the population of cattle wandering on the streets/roads/highways of Raipur city of India using the direct headcount method and advanced Photographic Capture-Recapture Method (PCRCM). We compared these two methods of population estimation to check their suitability and adequacy. We superimposed 163 grids (1.0 x 1.0 km each) on the map of Raipur city using Quantum Geographic Information System (QGIS) software. We randomly selected 20 grids for the estimation of the street cattle population. We used both line transect and block count sampling techniques under the direct headcount method. The estimates of visibly roaming cattle on the Raipur city streets were 11808.45 and 11198.30 using the former and the latter sampling techniques, respectively. Further, advanced PCRCM indicated an estimated 35149.61 and 34623.20 cattle using the line transect and block counting sampling techniques, respectively. We observed a female-biased sex ratio in both mature and immature cattle. The frequency of mature cattle was significantly higher than that of naive cattle, followed by the calf. Further, we noticed the frequency of cattle in a grid in the following order: cow > bull > heifer > immature male > female calf > male calf. We concluded that the estimated population of street cattle in Raipur city is about 35 thousand. The results of both the techniques, i.e., direct headcount method and PCRCM, are consistent for population estimation. The direct headcount method yields the number of cattle visibly roaming on the street at a particular time. In contrast, advanced PCRCM gives the total population of street cattle in the city. Active surveillance of the urban cattle population might be of critical importance for municipal and city planners. A better understanding of the urban cattle population might help mitigate the cattle menace on the street, eventually preventing cattle-human conflict and minimizing road accidents. The techniques adopted in this study will also help estimate the population of free-ranging dogs and other wildlife animals in any target location.

of the Doctor of Philosophy thesis of one of the
authors, BKS, who is getting a Junior Research
Fellowship under the scheme CSIR-UGC NET for
JRF [Sr. No. 2121530765. Ref. No: 20/12/2015(ii)
EU –V; dated 18/05/2016].

**Competing interests:** The authors have declared
that no competing interests exist.

## Introduction

Man began to domesticate cattle about 10000- to 8000-year ago [1, 2]. They used them as
sources of milk and meat. Cattle are distributed worldwide and constitute the most significant
number among the ungulates [3, 4]. In the animal world, cattle represent the largest zoomass
(about 600 million metric tons) [5].

At the end of 2018, the worldwide cattle population was about 996.36 million [3]. As per
the global cattle inventory 2018, India possesses the most significant number of cattle (i.e.,
305.00 million) in the world [4]. This figure was about 196.19 million during 2012 (19th live-
stock census, 2012) that included 5.29 million strays (free-roaming ownerless cattle) [6].
Recently, India's 20th livestock census (2019) was released. It showed that the cattle population
is about 198.48 million, including 5.02 million stray cattle in India [7]. In the India's State of
Chhattisgarh, the cattle population is about 9.98 million, including 0.374 million in urban
areas; and the stray cattle population is about 0.185 million [7].

As a consequence of the increasing cattle population, many cattle are wandering here and
there on streets in most developing and underdeveloped countries [8–11]. Rapid urbanization,
facilitated by both the government and private sectors, is attributed to increased street cattle.
Due to the shortage of foraging spots, cattle wander freely on the streets of urban areas. One of
the crucial reasons for the increased number of cattle on the urban street is space crunch; peo-
ple do not prefer to keep their domestic cattle at home. Owners release their cattle on the
nearby streets either temporarily or permanently.

People in urban areas share public places with bovine species. Both of them interact directly
or indirectly and face a lot of problems [8–11]. Due to the lack of a natural diet, cattle consume
harmful materials, like polythene and heavy metals [12–14]. They can neither digest nor egests
polythene/plastic materials. These toxic materials stick into their stomach that eventually
causes various pathological problems, such as indigestion, impaction, tympany, polybezoars,
immunosuppression [12, 13], and finally leads to early death [12–14]. About 95% to 100% of
street cattle ingest these hazardous materials into their stomach [12, 13]. These harmful mate-
rials and their metabolites slowly enter the human system through milk, milk products, and
meat leading to several health-related consequences.

Further, contamination of milk of urban cattle with traces of detergents, bleaching agents,
fertilizers, and cancer-causing agent 'dioxin' is well known [14, 15]. In India, people use cow/
cattle for milk, meat, agriculture, cart pulling, cow dung used for manure, plow, etc. Further,
the street cows are primarily used for milk. Usually, male street cattle do not have owners. In
contrast, a majority of the cows seen on the street have owners. However, space crunch around
the houses compels the owners to release their cows on the street either permanently or tempo-
rarily. But, whenever a cow gives birth to the calves and starts lactating, the respective owners
take them from the roads for milking purposes during morning and evening hours. After that,
again, the owners release the cows onto the streets. The city people let the dried cows and male
calves free on the roads permanently.

The presence of cattle on the street causes increased road accidents in urban areas [16]. Peo-
ple do not feel safe while walking or riding two/four-wheelers on the road. In the last 2.5 years
in the State of Punjab in India, nearly 300 people died on the street attributed to the stray cattle
menace [16].

Management of street cattle is one of the critical issues in many countries, such as India [9–11],
Ethiopia [13], Kenya [17, 18], Sri Lanka [19], and Nepal [20]. Management of these street cattle can
be effective only when we know sufficiently about their population and behavioral disposition.

Population estimation of cattle in India is carried out based on a door to door inventory
collected from the cattle owners [6, 7]. This procedure may not provide an accurate number of

cattle present on the street, especially in urban areas. People do not like to keep their cattle at home and mislead investigating officers by providing false information.

There are several methods to estimate the population of mammalian species. Capture Mark Recapture Method (CMRCM) is a very popular and widely accepted technique. In this technique, first, animals are captured and marked with either color [21–24] or some tags attached to the animal's body [25]. After a desirable time lag (in days), animals of the same area are recaptured. Based on the number of animals on two different days with the number of previously marked animals on the recapture day, the population is estimated. The estimation by the Lincoln index is considered useful in this context. Lincoln index is one of the most widely used and accepted indices in ecology to estimate any species' population. The color or physical tags used in the CMRCM for identifying individuals may alter the studied animals' physiology and behavior [25]. Sometimes, it may lead to the death of the animals. This fatality might be due to the hazardous nature of the colored chemical and its amount used for tagging. Another negative aspect of this method is that animals become vulnerable to their predators because of the marker elements.

In the advanced technique of CMRCM known as Photographic Capture-Recapture Method (PCRCM), marking animals with the hazardous colored chemicals or tag is not required. In this technique, the natural color pattern [26, 27] or other biographical and morphological characteristics [9, 28, 29] of the animals are necessary to identify the individuals without any direct/indirect harmful impact on them.

Another method used in recent studies is the direct headcount method [30, 31], which estimates density based on the number of heads per kilometer distance. It is very similar to the line transect sampling method [32]. Headcount sampling involves either block count sampling method or line transect sampling method [32]. These sampling methods determine the density of a given species in a unit area or the route's length. Subsequently, one can estimate the total population by multiplying the density with the entire area in the block count sampling method or multiplying with the routes' full length in the line transect sampling method [30, 32].

In the current study, we estimated the population of street cattle wandering on the streets/roads/highways of Raipur city and validated the direct headcount method's appropriateness by comparing our results with that emerged from the PCRCM method.

## Material and methods

### Study area and subject

The Raipur city (Latitude: 21.2514˚ N; Longitude: 81.6296˚ E, and Elevation: 298 m from the sea level) was our study's location. It is the largest city and Chhattisgarh's capital (a state of the Indian Federal Union). It has a human population of about 1.87 million [33]. The total length of streets/roads/highways of Raipur city is 1622.04 km distance. It transects over a 226 km$^2$ area of the city as per Municipal Corporation Raipur [34].

In the current study, we used the cattle as the subjects wandering, foraging, standing, or lying on the street of Raipur city. We considered all observed cattle on the streets/roads/highways, including 10 meters on both sides, as street cattle, irrespective of the age groups. Most of India's cattle are related to the indigenous species *Bos indicus* or the exotic species *Bos taurus*. The former includes Kosali, Sahiwal, Gir, Red Sindhi, Ongole, and Tharparkar, whereas the latter includes Holstein Friesian and Jersey [35].

### Preparation and selection of grid

We procured the map of Raipur city from the Municipal Corporation Raipur. We drew 190 grids (1.0 x 1.0 km each). We superimposed them on the map of Raipur city using Quantum

Geographic Information System (QGIS) software [36]. Of the 190 grids, we only included 163 grids for this study, reflecting at least 50% of the grid's locality. Further, we identified each grid by a Grid ID number from 1 to 163. We chose 20 out of 163 grids randomly for the sampling of the street cattle population. We generated 20 random numbers from among 1 to 163 by random number generation technique using Microsoft Excel.

## Sampling method

We carried out the study between February 16, 2019, and April 11, 2019. We followed line transect [30, 32] and block count methods [32] to sample the data. During sampling hours, two observers made direct observation of cattle. We followed all highways/roads/streets of the selected grids using the Maps Ruler Lite [37] mobile apps by walking and driving/riding a motorcycle. We sampled data on two different days. On the first day, we photographed the street cattle using a Nikon D-3400 still camera at the selected grid covering all the routes. On the third day, we retook photographs (recaptured) at the same grid approximately simultaneously. We repeated the same protocol for all 20 selected grids. We started and ended the street cattle photography from the predefined starting point to the sampling grid's endpoint, respectively. We followed the same route of the grid in a unidirectional way during both capturing and recapturing days. We ensured to examine each route but avoided repeated observation of the same route and double count. We noted down biographical (gender, age class), morphological (body color, horn shape & size, udder size), and additional characteristics (collar, bells, and cut mark in the ear) of each cattle on the spot during sampling hours (Table 1).

We carried out fieldwork between 11:00 h and 15:00 h to increase the chances of sighting the subject [27]. At this time, mostly street cattle are active and busy in foraging on the street or at the roadside open garbage and dumping bins [9, 11].

## Ethical approval statement

We photographed the animal subject from a distance. It does not necessitate any ethical approval. Our study protocol did not involve the handling of animals for any purpose. We did

**Table 1. Biographical, morphological, and additional characteristics/traits used for identification of individual street cattle in Raipur city using the advanced photographic capture-recapture method (PCRCM).**

| Characteristics/traits | Categories |
|---|---|
| Biographical | |
| Gender | Male/Female |
| Age class | Male: Bull/Immature male/Male calf |
| | Female: Cow/Heifer/Female calf |
| Morphological | |
| Color | Main body color |
| | Patch/ mark color |
| Udder size of a female | Largest/Larger/Large/Medium/Small/Smaller/Smallest |
| Presence of horn | Present/Absent |
| Size of horn | Largest/Larger/Large/Medium/Small/Smaller/Smallest |
| Direction of horn | Upward/Downward/Forward/Backward/Sided |
| Broken horn | Left horn/Right horn |
| Additional characteristics | |
| Presence of collar/rope | Present/Absent |
| | If present, the color of collar/rope |
| Presence of bell/rope in the collar | Present/Absent |
| Presence of ear-tag or cut-mark in ear | Present/Absent |

not harm the cattle directly or indirectly for any rationale of the study. We always maintained a 10 m distance from the animal subject during photography. The procedure did not disturb their natural behavior. Since we carried out this study in public places within the city, we obtained permission from the Municipal Corporation, Raipur, Chhattisgarh [Ref. number: 170/health department/MCR/2018; dated: 15/03/2018].

## Statistical analysis

**Estimation of population of street cattle.**  For estimating the population of street cattle, we used both the direct headcount method and Photographic Capture-Recapture Method (PCRCM). We did not find common cattle at Grid #89 (Jai Stambh Area) while recapturing. Therefore, we removed this grid from the dataset while using the PCRCM method. However, we used data from this area in the calculation of the direct headcount method.

*Method 1*: *Direct headcount method*. We observed street cattle visibly roaming on the street using two sampling methods, namely the line transect sampling method [30, 32] and the block count sampling method [32]. To estimate the total population, we first determined the density of street cattle.

We defined the density of cattle as outlined below:

The number of cattle per kilometer distance (for line transect sampling method):

$$\frac{\textbf{Total number of cattle observed during sampling in a grid}}{\textbf{Total distance covered in that grid during sampling without repetition of route}}$$

Subsequently, we estimated the *total number of cattle visibly roaming on the street* of Raipur city using the formula:

Total number of cattle = Average density × Total distance of street/road/highway in Raipur city

The number of cattle per square kilometer area (for block count sampling method):

$$\frac{\textbf{Total number of cattle observed}}{\textbf{Total area of sampled grid (in square km)}}$$

Total number of cattle = Average density × Total square km area of Raipur city

*Method 2*: *PCRCM method & Lincoln index formula*. Photographic Capture-Recapture Method (PCRCM) and Lincoln index formula were mutually used to estimate the street cattle population in 19 studied grids. We first estimated the population of street cattle in 19 different grids using the Lincoln index formula:

$$\frac{\textbf{n1}}{\textbf{N}} = \frac{\textbf{n3}}{\textbf{n2}}$$

$$\text{Thus, } \textbf{N} = \frac{\textbf{n1} \times \textbf{n2}}{\textbf{n3}}$$

Where,

n1 = number of cattle photographed on the first day;

n2 = number of cattle photographed on the third day;

n3 = number of cattle sighted on the first day and also on the third day;

N = Estimated population of cattle in a grid (measured in 1 km$^2$ area)

We also estimated the average number of street cattle in a unit square kilometer area (average density). After that, we calculated the *total population of street cattle in Raipur city* using the following formula:

The total population of Raipur = Average density × Total square kilometer area in Raipur city

*Other population ecological variables.* We further analyzed gender composition, sex ratio, and age composition of the street cattle.

We employed one way ANOVA and Mann-Whitney U test to compare the density of cattle in 20 grids, and the frequency of male and female cattle within each age group, respectively, at $p \leq 0.05$. Besides, we derived the sex ratio from the number of male street cattle divided by female street cattle.

We checked the normality and homogeneity of the data using the Kolmogorov-Smirnov statistic and Levene Statistic, respectively. We compared the frequency of cattle among three different age groups (mature, immature, and calf) and six other cattle groups (cow, bull, heifer, immature male, female calf, and male calf) using the non-parametric ANOVA (the Kruskal-Wallis rank test) followed by the Mann-Whitney U test. We used Bonferroni adjustment in both cases. We analyzed the data with the help of SPSS (version 20.0) [IBM SPSS Statistics for Windows, Version 20.0 (IBM Corp., Armonk, N.Y., USA)].

## Results

### Population density of cattle

**Method 1: Direct headcount method.** Table 2 represents the results of the direct head-count method. The total headcounts during the first and the third day were 998 and 984, respectively. The average number of street cattle was 991. The observed average density in 20 studied grids was 49.55 cattle $km^{-2}$ by the block count sampling method. Further, we covered 141.68 km in all 20 grids during capture on the first day and repeated the same route during recapturing on the third day. According to the line transect sampling method, the observed average number and average density of street cattle in all 20 grids were 145.53 and 7.28 $km^{-1}$ distance, respectively.

**Method 2: PCRCM method & Lincoln index formula.** We estimated the population of street cattle in 19 studied grids separately using the advanced PCRCM and Lincoln index (Table 3). The total estimated population of street cattle in all 19 grids was 3195.92. In the unbiased estimation, the estimated population was 2910.85 street cattle in 19 studied grids. The maximum estimated population, i.e., 367.50 street cattle, was found in Grid—27 (Bhan-puri). The minimum population, i.e., 14.67 street cattle, was estimated in Grid—36 (WRS Colony). Further, the estimated average density of street cattle was 168.21 per $km^2$ in 19 sampled grids. On the other hand, according to unbiased estimation, the estimated average density of street cattle was 153.20 per $km^2$.

### The estimated population size of street cattle in Raipur city

**Method 1: Direct headcount method.** According to the Municipal Corporation, Raipur, there is a 1622.04 km street/road/highway network in a 226 $km^2$ area in Raipur city. By the line transect sampling method, we estimated 11808.45 street cattle in Raipur city. By block count sampling method, the estimated street cattle in 226 $km^2$ area were 11198.30 (Table 4). Here, we derived both the estimated population without using the Lincoln index.

**Method 2: PCRCM method & Lincoln index formula.** The estimated population of street cattle in the total area (226 $km^2$) of Raipur city was 38015.46. On the other hand, according to unbiased estimation, the estimated population of street cattle in the city was 34623.20 (Table 4), based on the block count sampling method. Table 4 also revealed a similar output of the direct headcount method and PCRCM method. We found that the former method

**Table 2. The density of street cattle: Average number of cattle per km² area§ and km distance† in Raipur city-based on studies conducted at 20 randomly selected grids.**

| S. No. | Grid ID | GPS position | | Distance covered during each sampling days (km) | Number of cattle captured (Cattle per km² area) § | | Number of cattle captured (Cattle per km distance) † | | Density of cattle | |
|---|---|---|---|---|---|---|---|---|---|---|
| | | Latitude | Longitude | | First day | Third day | First day | Third day | The average number of cattle per km² area§ | The average number of cattle per km distance† |
| 1. | 83* | 21.2459 | 81.5784 | 5.37 | 79 | 62 | 14.71 | 11.55 | 70.5 | 13.13[ij] |
| 2. | 98 | 21.2380 | 81.5831 | 6.90 | 31 | 52 | 4.49 | 7.54 | 41.5 | 6.01[ef] |
| 3. | 101 | 21.2402 | 81.6087 | 4.39 | 94 | 113 | 21.41 | 25.74 | 103.5 | 23.58[k] |
| 4. | 36 | 21.2849 | 81.6493 | 5.69 | 11 | 8 | 1.93 | 1.41 | 9.5 | 1.67[abc] |
| 5. | 27 | 21.2887 | 81.6435 | 6.31 | 75 | 49 | 11.89 | 7.77 | 62 | 9.83[gh] |
| 6. | 110 | 21.2382 | 81.6915 | 5.67 | 20 | 17 | 3.53 | 3.00 | 18.5 | 3.26[abcde] |
| 7. | 94 | 21.2485 | 81.6809 | 3.24 | 19 | 32 | 5.86 | 9.88 | 25.5 | 7.87[fg] |
| 8. | 143 | 21.2131 | 81.6281 | 7.26 | 89 | 83 | 12.26 | 11.43 | 86 | 11.85[hij] |
| 9. | 115 | 21.2313 | 81.6274 | 8.57 | 122 | 122 | 14.24 | 14.24 | 122 | 14.24[j] |
| 10. | 103 | 21.2394 | 81.6288 | 8.63 | 52 | 50 | 6.03 | 5.79 | 51 | 5.91[ef] |
| 11. | 89 | 21.2486 | 81.6366 | 8.66 | 6 | 1 | 0.69 | 0.12 | 3.5 | 0.40[a] |
| 12. | 32 | 21.2838 | 81.6105 | 7.46 | 49 | 36 | 6.57 | 4.83 | 42.5 | 5.70[def] |
| 13. | 107 | 21.2378 | 81.6652 | 8.04 | 38 | 39 | 4.73 | 4.85 | 38.5 | 4.79[bcdef] |
| 14. | 76 | 21.2561 | 81.6560 | 10.30 | 51 | 47 | 4.95 | 4.56 | 49 | 4.76[bcdef] |
| 15. | 11 | 21.3109 | 81.6285 | 7.54 | 18 | 18 | 2.39 | 2.39 | 18 | 2.39[abcd] |
| 16. | 1 | 21.3289 | 81.6285 | 7.12 | 11 | 12 | 1.54 | 1.69 | 11.5 | 1.62[ab] |
| 17. | 59 | 21.2755 | 81.6372 | 6.57 | 33 | 32 | 5.02 | 4.87 | 32.5 | 4.95[cdef] |
| 18. | 135 | 21.2199 | 81.6751 | 9.02 | 66 | 78 | 7.32 | 8.65 | 72 | 7.98[fg] |
| 19. | 34 | 21.2846 | 81.6281 | 4.33 | 20 | 24 | 4.62 | 5.54 | 22 | 5.08[def] |
| 20. | 70 | 21.2568 | 81.6001 | 10.61 | 114 | 109 | 10.74 | 10.27 | 111.5 | 10.51[ghi] |
| Total | | | | 141.68 | 998 | 984 | 144.92 | 146.12 | 991 | 145.53 |
| Average density in 20 studied grids | | | | | | | | | 49.55 | 7.28 |

Direct headcount using

§Block count sampling method;

†Line transect sampling method; Means bearing the identical superscripted letters are not statistically significantly different from each other (Based on Duncan's multiple range test).

*[83]Sarona; [98]Indraprasth Colony; [101]Daganiya Colony; [36]WRS Colony; [27]Bhanpuri; [110]Labhandi Colony; [94]Jivan Vihar; [143]Bhatagaon; [115]Purani Basti; [103]Amin Para; [89]Jai Stambh Area; [32]Sondongari; [107]Telibandha; [76]Shankar Nagar; [11]Kailash Nagar; [1]Transport Nagar; [59]Shivanand Nagar; [135]Mahaveer Nagar; [34]Govardhan Nagar; [70]Kota Colony.

underestimated the population of the cattle visibly roaming on the street. In contrast, the latter method estimated the population of street cattle in the whole city.

## Results of other population ecological variables

**Effect of the factor "location" on the density of street cattle.** Cattle per km in different studied grid was significantly different ($F_{19,20}$ = 30.24; $p < 0.001$). As per the line transect sampling method, the average number of cattle per km distance was the maximum (23.58 cattle per km) at Grid—101 (Daganiya Colony) and the minimum (0.40 cattle per km) at Grid—89 (Jai Stambh Area). On the other hand, as per the block count sampling method, we found the maximum density (122 cattle per km²) at Grid—115 (Purani Basti) and the minimum density (3.5 cattle per km²) at Grid—89 (Jai Stambh Area) (Table 2).

**Table 3. The estimated population of street cattle per km square area using advanced photographic capture-recapture method (PCRCM) & Lincoln index at 20 different sampling grids in Raipur city.**

| S. No. | Grid Number | Capturing Day (CD) | Recapturing Day (RCD) | Day 1 (cattle per km²) n1 | Day 3 (cattle per km²) n2 | Common in day 1 and day 3 (cattle per km²) n3 | Estimated cattle population in per km² area N = n1×n2/n3 |
|---|---|---|---|---|---|---|---|
| 1. | 83 | 14/02/2019 | 16/02/2019 | 79 | 62 | 20 | 244.90 |
| 2. | 98 | 18/02/2019 | 20/02/2019 | 31 | 52 | 12 | 134.33 |
| 3. | 101 | 22/02/2019 | 24/02/2019 | 94 | 113 | 39 | 272.36 |
| 4. | 36 | 25/02/2019 | 27/02/2019 | 11 | 8 | 6 | 14.67 |
| 5. | 27 | 25/02/2019 | 27/02/2019 | 75 | 49 | 10 | 367.50 |
| 6. | 110 | 26/02/2019 | 28/02/2019 | 20 | 17 | 6 | 56.67 |
| 7. | 94 | 09/03/2019 | 11/03/2019 | 19 | 32 | 4 | 152.00 |
| 8. | 143 | 12/03/2019 | 14/03/2019 | 89 | 83 | 25 | 295.48 |
| 9. | 115 | 13/03/2019 | 15/03/2019 | 122 | 122 | 52 | 286.23 |
| 10. | 103 | 16/03/2019 | 18/03/2019 | 52 | 50 | 21 | 123.81 |
| 11. | 89* | 19/03/2019 | 21/03/2019 | 6* | 1* | 0* | undefined* |
| 12. | 32 | 20/03/2019 | 22/03/2019 | 49 | 36 | 12 | 147.00 |
| 13. | 107 | 23/03/2019 | 25/03/2019 | 38 | 39 | 20 | 74.10 |
| 14. | 76 | 26/03/2019 | 28/03/2019 | 51 | 47 | 11 | 217.91 |
| 15. | 11 | 27/03/2019 | 29/03/2019 | 18 | 18 | 4 | 81.00 |
| 16. | 1 | 30/03/2019 | 01/04/2019 | 11 | 12 | 2 | 66.00 |
| 17. | 59 | 03/04/2019 | 05/04/2019 | 33 | 32 | 7 | 150.86 |
| 18. | 135 | 04/04/2019 | 06/04/2019 | 66 | 78 | 29 | 177.52 |
| 19. | 34 | 08/04/2019 | 10/04/2019 | 20 | 24 | 6 | 80.00 |
| 20. | 70 | 09/04/2019 | 11/04/2019 | 114 | 109 | 49 | 253.59 |
| Total estimated population in 19 km² area | | | | | | | **3195.92** |
| Average density | | | | | | | **168.21** |
| Unbiased estimated population in 19 km² area | | | | 992 | 983 | 335 | **2910.85[a]** |
| Unbiased average density | | | | | | | **153.20** |

n1Number of cattle photographed on the first day; n2Number of cattle photographed on the third day; n3Number of cattle sighted on the first day and also present on the third day; NEstimated total population of cattle in 19 grid (1.0 x 1.0 km);

*grid number 89 was not included in the calculation because the number of street cattle was undefined as per the Lincoln index formula when common street cattle in two sampling days were zero,

[a]unbiased estimation.

**Gender composition.** Fig 1 depicts the results of the Mann-Whitney U test. Results indicate that the overall number of sighted female cattle was significantly higher than that of the male cattle ($U = 431.50$; $p < 0.001$). We also compared the frequency of male and female cattle

**Table 4. The estimated total population of street cattle in Raipur city using two recommended methods, namely line transect sampling and block count sampling.**

| Methods used for population estimation | | Line transect sampling | | | Block count sampling | | |
|---|---|---|---|---|---|---|---|
| | | Density (number of cattle per km distance) | The total length of the road network in Raipur city (in km) | Estimated population in Raipur city | Density (number of cattle per km² area) | The total area of Raipur city (in km²) | Estimated population in Raipur city |
| Direct head count method | | 7.28 | 1622.04 | 11808.45 | 49.55 | 226 | 11198.30 |
| PCRCM and Lincoln index | Density | 24.38* | 1622.04 | 39545.34 | 168.21 | 226 | 38015.46 |
| | Unbiased density | 21.67* | 1622.04 | 35149.61 | 153.20 | 226 | 34623.20 |

*Refer S1 Table.

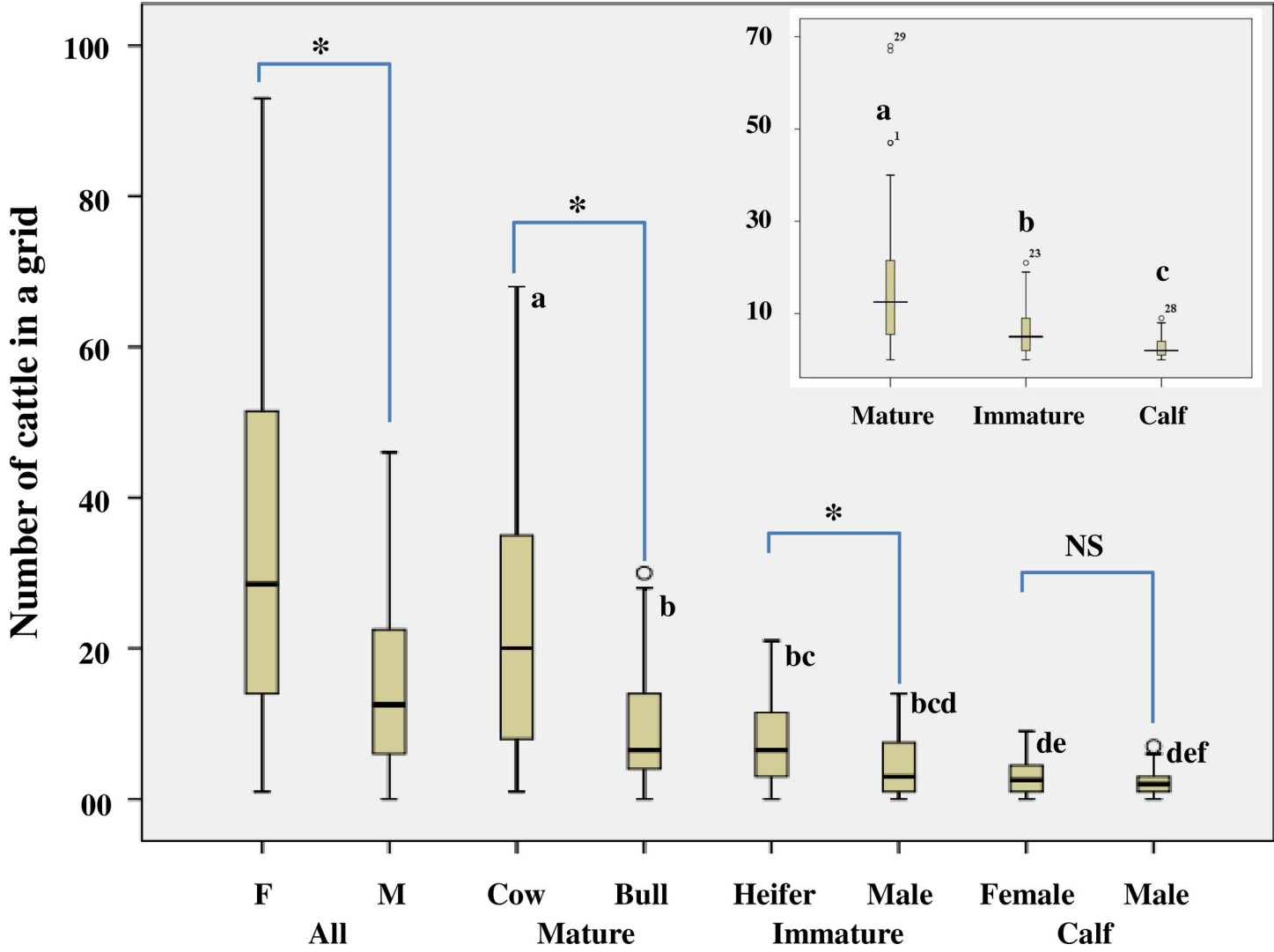

**Fig 1. Box plots depicting the number of cattle in a grid by age groups and by gender.** The horizontal line that divides the box into two parts represents the median value. The upper and lower limits of the box represent the 75th and 25th percentile. The top and the lower vertical lines outside the box indicate the maximum and the minimum values in the dataset. The unfilled circles outside the box denote potential outliers. The figure in the inset depicts the number of cattle in a grid as a function of age. *Indicates a statistically significant difference between male and female groups ($p \leq 0.05$). NSNot significant. Boxes bearing the identical superscripted letters are not statistically significantly different from each other [a significant level was set at $p \leq 0.017$ while comparing pairs among three age groups (Fig 1 inset), and $p \leq 0.003$ while comparing pairs among six cattle groups]. We have used these superscripted letters to show the difference only among the groups.

within different age groups. The number of male cattle was significantly lower than female cattle in two age groups, i.e., in mature group (U = 379.50; $p < 0.001$) and immature group (U = 508.00; $p < 0.05$). However, this difference was not significant in the calf group (U = 652.00; $p > 0.05$).

**Sex ratio.** The estimated sex ratio was 0.484:1 (male: female) based on 20 sampling grids. For each female, there was less than one male in all 20 studied grids. The maximum (0.75:1) and the minimum (0.118:1) sex ratios were in Jai Stambh Area and WRS Colony, respectively (S2 Table).

**Age composition.** We compared different age groups of the street cattle using the Kruskal-Wallis test. Before employing this test, we examined the normality (by Kolmogorov-Smirnov statistic) and homogeneity (by Levene statistic) of data. Results show that data were

neither normally distributed ($p < 0.05$) nor homogenous ($p < 0.05$). This result did not fulfill the parametric test's assumption; therefore, we performed the Kruskal-Wallis test. We considered the mean difference significant at $p \leq 0.05$. Results show a considerable difference in the number of sighted cattle among different age groups ($H_2 = 76.954$; $p < 0.001$). The number of street cattle was in the order of mature > immature > calf (Fig 1 inset). To know the differences between the different combinations of groups, we analyzed the data using the Mann-Whitney U test. Here the mean difference was set to be significant at a level of $p \leq 0.017$. We calculated this $p$-value using the Bonferroni adjustment. Initially, we divided the significance level used for overall mean differences (0.05) by the number of possible pairs in 3 groups (3 in this case). Subsequently, we had a new significance level, i.e., 0.017 (0.05/3). Results indicate that the number of cattle in the mature group was significantly higher as compared to that of the immature (U = 1773.00; $p < 0.001$) and calf groups (U = 797.50; $p < 0.001$). The frequency of cattle in the immature group was also significantly higher than the frequency of cattle in the calf group (U = 1753.50; $p < 0.001$) (Fig 1 inset).

For the frequency of six different cattle groups, data were not normally distributed ($p < 0.05$) except for the distribution of heifers ($p > 0.05$). Data were also found to be non-homogenous ($p < 0.05$). The result of the Kruskal-Wallis test shows that there was significant difference ($p < 0.05$) in the frequency of different age groups of cattle ($H_5 = 95.283$; $p < 0.001$) (Fig 1). Further, to know the differences among different combination of groups, data were analyzed using the Mann-Whitney U test. The significance level was set at $p \leq 0.003$ (we found this value by Bonferani adjustment, i.e., 0.05/15). There were 15 possible pairs in six different cattle groups. The frequency of cows in 1 km$^2$ grid was significantly higher than all other cattle groups (cow *vs* bull: U = 379.50; $p < 0.001$; cow *vs* heifer: U = 327.00; $p < 0.001$; cow *vs* immature male: U = 191.00; $p < 0.001$; cow *vs* female calf: U = 97.50; $p < 0.001$; cow *vs* male calf: U = 65.50; $p < 0.001$). The frequency of bulls was significantly higher as compared to that of the female calf (U = 355.50; $p < 0.001$), and male calf (U = 279.00; $p < 0.001$), but did not differ significantly with that of the heifers (U = 755.00; $p = 0.664$) and immature male cattle (U = 500.00; $p = 0.004$). Further, the frequency of heifers was significantly greater as compared to that of the female calf (U = 339.50; $p < 0.001$) and male calf (U = 243.50; $p < 0.001$), but did not find to be differ significantly with that of the immature male cattle (U = 508.00; $p = 0.005$). The frequency of immature male cattle was not found to be significantly differ with the frequency of male calf (U = 527.50; $p = 0.008$) and female calf (U = 643.00; $p = 0.127$). The frequency of female and male calf did not differ significantly (U = 652.00; $p = 0.148$) (Fig 1).

## Discussion

In India, people share public places with street cattle and street dogs. The number of cattle and dogs is continuously increasing on the street/road/highways of India. There are several studies regarding free-ranging urban dogs and their population management program in many countries [24, 27, 38–41]. However, there is a lack of information on the ecological and behavioral aspects of street cattle. We found only two publications while searching on the SCOPUS database [9, 11]. Both studies highlighted circadian rhythms in various behavioral variables of street cattle. One of these studies also estimated the population of street cattle in 12 sampling sites in one of the Indian cities (Sambalpur) but did not define the studied areas [9].

The human population is 1.87 million in Raipur city [33], and the estimated population of street cattle is 34623. Based on this finding, we can interpret that for every 54 people in the city, there is at least 1 street cow. This human: cattle ratio is an alarming and challenging situation, especially for the Raipur Municipal Corporation. The number of street cattle is on the

rise making the chances of man-cattle conflict higher. Many vehicular accidents occur every day due to cattle on the street [16], and even cattle also get injured [42]. Unfortunately, we could not find any data regarding the human-cattle conflict on the road in Raipur city. The State Government and Municipal Corporation, Raipur, have initiated work to discover ways to mitigate this cattle menace.

In this study, we observed and evaluated the density and population of street cattle using two different scientific survey protocols. Estimating the population using the density of direct headcount method in terms of animal per km distance is a novel method. In two recent studies, the authors used this method to estimate street dogs' density in different countries in cohort studies [30, 31]. The authors evaluated street dogs' density in terms of the number of street dogs per km distance. A recent study suggested that it is possible to estimate the street dog population by multiplying the density by the road's entire length in a particular area where the population is determined [30]. They also recommended conducting comparable studies to validate this novel method.

Advanced PCRCM is a well established and appropriate method. It has been successfully used in different studies to estimate other species' populations [21–24, 26, 27, 41]. Most recently, Arya and colleagues used the method to determine the population of street cattle in 12 study sites of Sambalpur city, Odisha, India, and the authors recorded about 146 street cattle on the 12 studied sites [9]. They neither defined the size of the area nor estimated the total population of cattle in the city. This study used 20 randomly chosen 1 km$^2$ sampling grids and computed the total estimated population of street cattle in Raipur city.

We found that the estimated population of cattle on the street of Raipur city was about 11808 (by line transect sampling) and 11198 (by block count sampling) using the direct headcount method. The estimated total population of street cattle in Raipur city was 35150 (by line transect sampling) and 34623.20 (by block count sampling) based on the advanced Photographic Capture-Recapture Method (PCRCM) & Lincoln index formula (Table 4). The estimated population of the direct headcount method always remains lower as compared to the PCRCM. The possible reasons for lower estimation by direct headcount are: (a) this method computed the average number of street cattle on two different days, (b) it does not include the cattle not visible within the observational window, (c) some cattle were not present on the third day during recapture and could not be photographed, and (d) we only counted the cattle that were seen on the street/road/highway and within adjacent 10 meters on both sides during our observation. However, the PCRCM estimates the total population of street cattle found in the city irrespective of their presence or absence during two observational days.

We can state that the direct headcount method only estimates the average number of heads visible on the street at a given time. In contrast, the PCRCM estimates the total population of street cattle found in the city irrespective of their presence or absence during two observational days. The direct headcount method appears to be adequate and appropriate for this and possibly for other comparable studies. This method is an effective technique to estimate the average population of any species on a particular line transect at a given time. However, if someone is interested in determining any species' total population, we recommend the Lincoln index formula.

A model diagram depicts the primary difference between the above two methods for ease of understanding (Fig 2). Calculated density and estimated cattle population are presented in Fig 2A and 2B by direct headcount method and Fig 2C and 2D by PCRCM.

Based on data emanated from our studies of 20 randomly chosen sampling grids, the average population density of street cattle in Raipur city was 7.28/km. It varied from 0.40/km (min) to 23.58/km (max) (Table 2). The density depends on the extent of urbanization in the area. In our study, we found a minimum density of street cattle in the Jai Stambh Area. It is one of the most popular areas of the city, having fewer residential households (3100) [34] and

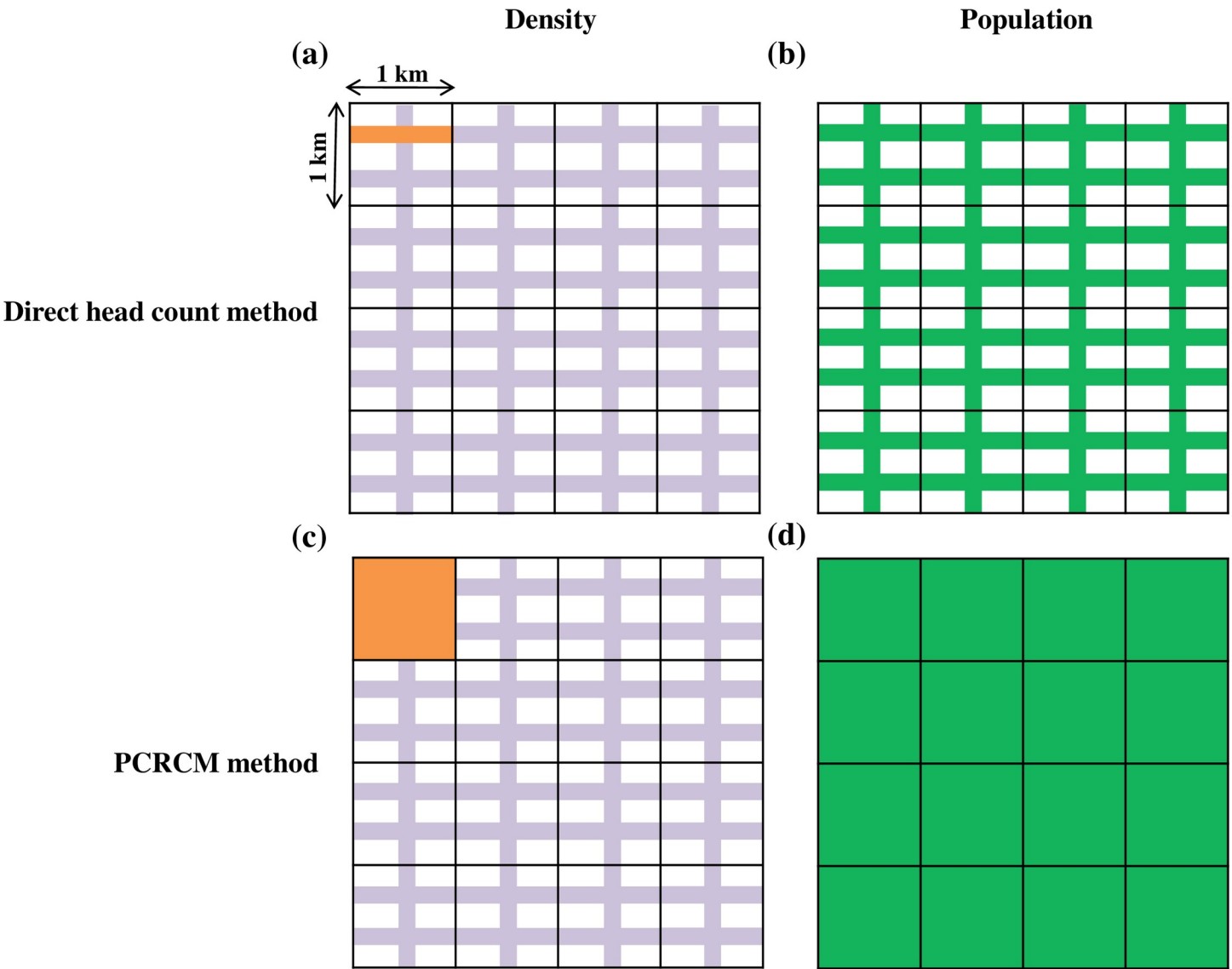

**Fig 2. Model diagram—density and population estimation of street cattle by two methods used in the study.** (a) Density is represented (orange color) in terms of the number of cattle per km distance and (b) estimated population in total road network (green color) by direct headcount method. (c) Density (orange color) in terms of the number of cattle per km² area and (d) estimated population in total area (green color) by PCRCM. The light purple color represents the road network.

mainly used by people for trade and business purposes. This area becomes mostly over-crowded between 0900 to 2300 h. The area is situated at the center of the city and has no open garbage disposal places. It may be one possible reason for a lower density of the street cattle in this area. There is minimal opportunity for the cattle to find food there.

On the other hand, the Daganiya area had a maximum density of street cattle. People occupied this area for a residential purpose (7075 households) [34]. In this location, cattle easily find food materials because this area has many open garbage disposal places.

Further, some earlier studies in natural environments have reported a male-biased sex ratio in the cattle population [43, 44]. In contrast, we found a female-biased sex ratio of street cattle (male: female = 0.48:1). In the current study, we observed a higher frequency of females in all studied grids (S2 Table) and at different age levels (Fig 1).

Age composition analysis revealed that mature street cattle form 64.44% of the total population (S1 Fig). They have a high breeding potential. They contribute significantly to the cattle's population growth on the streets of the city. However, this hypothesis needs to be validated and confirmed by executing studies in the future, employing the same population estimation techniques.

## Conclusions

The current study fell in the realms of urban ecology. It revealed a staggeringly high number of cattle on the street of Raipur city, using two different methods, namely, direct headcount method and PCRCM. The former is suitable for point estimation of cattle population at a given time. Simultaneously, the latter is the apt method for assessing the global cattle population of a designated city. Similar future studies in other urban spaces are likely to validate the procedures used in this study. Active surveillance of the urban cattle population might be of critical importance for municipal and city planners. A better understanding of the urban cattle population might help mitigate the cattle menace on the street, eventually preventing cattle-human conflict and minimizing road accidents. The techniques adopted in this study will also help estimate the population of free-ranging dogs and other wildlife animals in any target location.

## Supporting information

**S1 Fig. Age composition (in percentage) in the population of street cattle in Raipur city.**
(TIF)

**S1 Table. Estimated population of street cattle per km distance after applying the data based on line transect sampling into the Lincoln index formula at 20 different sampling grids in Raipur city.**
(PDF)

**S2 Table. Sex ratio of street cattle in different studied grid: Sex ratio is the ratio of males to females in a population.**
(PDF)

## Acknowledgments

We are thankful to the Municipal Corporation of Raipur, India, to provide us with the data on garbage bins installation and a map of Raipur city. We record our special thanks to Mr. Radhe Lal Markam for assisting us during the field study. We are thankful to Dr. Anjali Tripathy, Associate Professor at the School of English, Gangadhar Meher University, Sambalpur, Odisha, for the manuscript's language editing. We are also obliged to two anonymous reviewers for offering us valuable suggestions for the improvement of the manuscript.

## Author Contributions

**Conceptualization:** Bhupendra Kumar Sahu, Arti Parganiha, Atanu Kumar Pati.

**Data curation:** Bhupendra Kumar Sahu.

**Formal analysis:** Bhupendra Kumar Sahu, Arti Parganiha.

**Investigation:** Bhupendra Kumar Sahu.

**Methodology:** Bhupendra Kumar Sahu, Arti Parganiha, Atanu Kumar Pati.

**Supervision:** Arti Parganiha, Atanu Kumar Pati.

**Validation:** Arti Parganiha, Atanu Kumar Pati.

**Visualization:** Bhupendra Kumar Sahu, Arti Parganiha, Atanu Kumar Pati.

**Writing – original draft:** Bhupendra Kumar Sahu.

**Writing – review & editing:** Bhupendra Kumar Sahu, Arti Parganiha, Atanu Kumar Pati.

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
