## [Decision Letter · Decision Letter 0]

5 Aug 2020

PONE-D-20-16241

Population estimation study reveals a staggeringly high number of cattle on the streets of urban Raipur in India

PLOS ONE

Dear Dr. Parganiha,

Thank you for submitting your manuscript to PLOS ONE. After careful consideration, we feel that it has merit but does not fully meet PLOS ONE’s publication criteria as it currently stands. Therefore, we invite you to submit a revised version of the manuscript that addresses the points raised during the review process.

Your manuscript was reviewed by two experts in the field, and they have recommended some modifications be made prior to acceptance.

I therefore invite you to modify your manuscript and to resubmit it.

If you could write a response to reviewers, that will expedite review upon resubmission.

I wish you the best of luck with your revisions.

Hope you are keeping safe and well in these difficult times.

We look forward to receiving your revised manuscript.

Kind regards,

Simon Clegg, PhD

Academic Editor

PLOS ONE

"He obtained a fellowship from the University Grants Commission, New Delhi, India, under the

scheme of Joint CSIR-UGC NET for Junior Research Fellowship [Sr. No. 2121530765. Ref.

No: 20/12/2015 (ii) EU –V dated 18/05/2016]."

3. We note that Figure 1 in your submission contain map images which may be copyrighted. All PLOS content is published under the Creative Commons Attribution License (CC BY 4.0), which means that the manuscript, images, and Supporting Information files will be freely available online, and any third party is permitted to access, download, copy, distribute, and use these materials in any way, even commercially, with proper attribution. For these reasons, we cannot publish previously copyrighted maps or satellite images created using proprietary data, such as Google software (Google Maps, Street View, and Earth). For more information, see our copyright guidelines: http://journals.plos.org/plosone/s/licenses-and-copyright.

3.1.    You may seek permission from the original copyright holder of Figure 1 to publish the content specifically under the CC BY 4.0 license.

3.2.    If you are unable to obtain permission from the original copyright holder to publish these figures under the CC BY 4.0 license or if the copyright holder’s requirements are incompatible with the CC BY 4.0 license, please either i) remove the figure or ii) supply a replacement figure that complies with the CC BY 4.0 license. Please check copyright information on all replacement figures and update the figure caption with source information. If applicable, please specify in the figure caption text when a figure is similar but not identical to the original image and is therefore for illustrative purposes only.

4. We note that Figure 2 includes an image of a participant in the study. 

Reviewers' comments:

Reviewer's Responses to Questions

**Comments to the Author**

1. Is the manuscript technically sound, and do the data support the conclusions?

Reviewer #1: Yes

Reviewer #2: Yes

2. Has the statistical analysis been performed appropriately and rigorously? 

Reviewer #1: Yes

Reviewer #2: Yes

3. Have the authors made all data underlying the findings in their manuscript fully available?

Reviewer #1: Yes

Reviewer #2: Yes

4. Is the manuscript presented in an intelligible fashion and written in standard English?

Reviewer #1: Yes

Reviewer #2: Yes

5. Review Comments to the Author

Reviewer #1: Dear Author,

I have read your submission with high interest. Even the topic is quite exotic for majority of readers, because not having free/wandering cattle on streets. Beside this I see nice implications in the methodology to be used in counting/ estimation of population size in free range/ wild animal populations where direct head counting is considered as very rough.

This is also one of my recommendation for conclusions to have wider view on application of the methodology not only in urban centres and domesticated animals.

It would be further interesting to have more detailed information on usage/ performance of street cattle in context of information given esp. on L.82.

L.86 reference missing... which countries?

...Cattle wandering, foraging, standing, or lying on the street of the city of Raipur were the subjects in the current study...can you give detailed description of cattle (breed/strain/ av.performance...)

...Management of the free-roaming cattle for making urban roads free from cows is beneficial to both the animals and the citizens....This is the first mentioned place with management...

From my point of view are the conclusions weakest part of the submission, clear explanation of the purpose of the study...

Even...The direct headcount method that we adopted in this study is a novel technique, and it always underestimates the population as compared to the PCRCM.....sounds strange, i.e.your study doesn't have sense? Also further implication ....can also be useful for estimating the community of other animal species...is very general without deeper support by the study.

Therefore I would ask you to review outcomes of your study to give implications/ clear message.

Reviewer #2: This is an interesting study which highlights the issue of the high numbers of cattle on the streets of India. It is certainly something which I remember from when I visited. The manuscript is generally well written, clear and well thought through. The conclusions however let it down as this is more of a summary than a conclusion.

I have written in some of the suggested minor changes below, which are largely typographical and grammatical. I have also made the changes which are required to allow the authors to just transfer it over.

It was a nice paper to read, so I commend you for your hard work and interesting research.

Line 57-58- this sentence seems incomplete. Represents the maximum what?

Line 59- As per the global cattle inventory (add in word)

Line 63- showed that the cattle population is about (add in word)

Line 67- As a consequence of the increasing cattle population (add in word)

Line 69- The increase in the number of street cattle can be attributed (add in word)

Line 76- Due to the lack of availability of a natural diet (add in word)

Line 81- not sure that sip is the correct word- maybe slip or seep

Line 82- do you have any data on cattle performance, usage or general health based on what you have said. Is there any breed data too as this would be interesting, or any data on sex?

Line 84- or riding two/four-wheelers (make plural)

Line 86- it would be nice to see this referenced with different countries

Line 89- mostly based on a door to door inventory (add in word)

Line 99- The Lincoln index is one of the most widely used (add in words)

Line 115- One can estimate the total population by multiplying the density 116 to the entire area or distance/length of the routes – this is unclear and I cant help to reword it as it doesn’t make sesne

Line 125- transects over a 226 km2 area of the city (add in word)

Line 129 -and within an adjacent 10 meters on both sides (add in word)

Line 134- Raipur city to find out the routes of highways/roads/streets (make plural)

Line 136-139- I found this difficult to understand. Please reword- In this study, we included the grids that covered ≥ 50% of the area (i.e., 163 grids: 151 grids covered 75-100% and 12 grids covered 50-75%). We excluded the remaining 27 grids (i.e., the grids those covered < 50% of area).

Line 153- highways/roads/streets-(make all plural)

Line 156- using a Nikon D-3400 still camera (add in word)

Line 163- was this data noted for all cattle seen?

Table one- calf not calve

Line 216- average number of street cattle in a unit square kilometer area (add in word)

Line 222-224 could be combined to aid reading flow

Line 228- calf rather than calve

Line 231- add in manufacturer for SPSS

Line 262- in the city

Line 268-269- Cattle per km in the different studied grids was significantly different (reword)

Line 281- calf rather than calve

Line 292- For each female, there was less than one male in all 20 studied grids. (reword)

Why were different values set for significant for the p values?

Line 306- For this, initially, the significant level used for overall mean differences (reword)

Line 308- significance level

Line 315- The result of the Kruskal-Wallis test shows that there was significant (reword)

Line 318- The significant level (reword)

Line 319- The frequency of cows in 1 km2 grid (make plural)

Line 319- you confused me a bit here by using cows compared to the other group. Can you define what you mean by a cow, and how you identified one? You have previously used cows to encompass all of the age groups

Line 322- The frequency of a bull was (add in word)

Line 325- Further, the frequency of heifers (add in word and make plural)

Line 343- Based on this finding, we can interpret that for every 54 people in the city, there is at least 1 street cow. (reword)

Line 368- line and block do not need capitalising- and throughout

Line 402- having a lower number of residential households (3100) (reword and make plural)

Line 405- possible reasons for a lower density of the street cattle (reword)

Line 408- On the other hand, the Daganiya area had the maximum density of street cattle. (reword)

Line 418- They have a high potential for breeding (reword)

Your conclusion isn’t a conclusion, more a summary. Please write an appropriate conclusion which may deal with control of the numbers etc. Is there any risk posed for disease transmission?

6. PLOS authors have the option to publish the peer review history of their article (what does this mean?). If published, this will include your full peer review and any attached files.

Reviewer #1: No

Reviewer #2: No

---

## [Author Response · Author response to Decision Letter 0]

17 Sep 2020

Action taken on suggested additional requirements

Point #1:

Action: Our revised manuscript followed the PLOS ONE's style requirements.

Point #2:

Thank you for stating the following in the Acknowledgments Section of your manuscript:

"He obtained a fellowship from the University Grants Commission, New Delhi, India, under the scheme of Joint CSIR-UGC NET for Junior Research Fellowship [Sr. No. 2121530765. Ref. No: 20/12/2015 (ii) EU –V dated 18/05/2016]."

Action: We have revised the Acknowledgments Section. We have removed funding-related text from the manuscript.

Please add the following in Funding Section.

Funding: The author(s) did not receive any specific funding for this work. However, this work is a part of the Doctor of Philosophy thesis of one of the authors, BKS, who is getting a Junior Research Fellowship under the scheme CSIR-UGC NET for JRF [Sr. No. 2121530765. Ref. No: 20/12/2015(ii)EU –V; dated 18/05/2016].

Point #3:

We note that Figure 1 in your submission contain map images which may be copyrighted. All PLOS content is published under the Creative Commons Attribution License (CC BY 4.0), which means that the manuscript, images, and Supporting Information files will be freely available online, and any third party is permitted to access, download, copy, distribute, and use these materials in any way, even commercially, with proper attribution. For these reasons, we cannot publish previously copyrighted maps or satellite images created using proprietary data, such as Google software (Google Maps, Street View, and Earth). For more information, see our copyright guidelines: http://journals.plos.org/plosone/s/licenses-and-copyright.

Action: We removed Figure 1 from our revised submission.

Point #4:

We note that Figure 2 includes an image of a participant in the study. 

Action: We removed Figure 2 from our revised submission.

Response to Reviewer(s)' Comments

We would like to express our gratitude towards the esteemed reviewers for providing useful suggestions. We tried our best to revise the manuscript in compliance with the valuable tips offered to us.

Reviewer #1

Reviewer comment 1: I have read your submission with high interest. Even the topic is quite exotic for majority of readers, because not having free/wandering cattle on streets. Beside this I see nice implications in the methodology to be used in counting/ estimation of population size in free range/ wild animal populations where direct head counting is considered as very rough.

Author response 1: We express our special thanks to the honorable reviewer for the compliments and appreciation of the manuscript.

Reviewer comment 2: This is also one of my recommendation for conclusions to have wider view on application of the methodology not only in urban centres and domesticated animals.

Author response 2: We have modified the section 'conclusions' in the revised manuscript. 

Reviewer comment 3: It would be further interesting to have more detailed information on usage/ performance of street cattle in context of information given esp. on L.82.

Author response 3: In India, people use cow/cattle for milk, meat, agriculture, cart pulling, cow dung used for manure, plow, etc. Further, the street cows are used mainly for milk. Usually, male street cattle do not have owners. In contrast, a majority of cows seen on the street have owners. However, space crunch around the houses compels the owners to release their cows on the street either permanently or temporarily. But, whenever a cow gives birth to the calves and starts lactating, the respective owners take them from the roads for milking purposes during morning and evening hours. After that, again, the owners release the cows onto the streets. The city people let the dried cows and male calves free on the roads permanently.

We have added the above phrases to the main revised text.

Reviewer comment 4: L.86 reference missing... which countries?

Author response 4: We have added names of the counties in the main revised text as "countries such as India, Ethiopia, Kenya, Shri Lanka, and Nepal” and have given the references.

Reviewer comment 5: ...Cattle wandering, foraging, standing, or lying on the street of the city of Raipur were the subjects in the current study...can you give detailed description of cattle (breed/strain/ av.performance...)

Author response 5: Most of the cattle present in India are related to the indigenous species Bos indicus or the exotic species Bos taurus. The former includes breeds, namely Kosali, Sahiwal, Gir, Red Sindhi, Ongole, Tharparkar, whereas the latter includes Holstein Friesian and Jersey. 

We did not study the performance of street cattle.

Reviewer comment 6: ...Management of the free-roaming cattle for making urban roads free from cows is beneficial to both the animals and the citizens....This is the first mentioned place with management...

Author response 6: Yes, it is. We are pleased to say thanks for the comment of the honorable reviewer. 

Reviewer comment 7: From my point of view are the conclusions weakest part of the submission, clear explanation of the purpose of the study...

Author response 7: We have modified the section ‘conclusions' in the revised manuscript.

Reviewer comment 8: Even...The direct headcount method that we adopted in this study is a novel technique, and it always underestimates the population as compared to the PCRCM.....sounds strange, i.e.your study doesn't have sense? Also further implication ....can also be useful for estimating the community of other animal species...is very general without deeper support by the study.

Therefore I would ask you to review outcomes of your study to give implications/ clear message.

Author response 8: We have modified the section ‘conclusions' in the revised manuscript.

Reviewer # 2

Reviewer comment 9: This is an interesting study which highlights the issue of the high numbers of cattle on the streets of India. It is certainly something which I remember from when I visited. The manuscript is generally well written, clear and well thought through.

Author response 9: We express our special thanks to the honorable reviewer to appreciate our scientific efforts in this manuscript.

Reviewer comment 10: The conclusions however let it down as this is more of a summary than a conclusion.

Author response 10: We have modified the section ‘conclusions' in the revised manuscript.

Reviewer comment 11: I have written in some of the suggested minor changes below, which are largely typographical and grammatical. I have also made the changes which are required to allow the authors to just transfer it over.

It was a nice paper to read, so I commend you for your hard work and interesting research.

Author response 11: We express our gratitude towards the esteemed reviewer for the useful suggestions to overcome the typographical and grammatical errors throughout the manuscript.

Reviewer comment 12: Line 57-58- this sentence seems incomplete. Represents the maximum what?

Author response 12: We have modified the statement as "In the animal world, cattle represent the maximum zoomass."

Reviewer comment 13: Line 59- As per the global cattle inventory (add in word)

Author response 13: Corrections made

Reviewer comment 14: Line 63- showed that the cattle population is about (add in word)

Author response 14: Corrections made

Reviewer comment 15: Line 67- As a consequence of the increasing cattle population (add in word)

Author response 15: Corrections made

Reviewer comment 16: Line 69- The increase in the number of street cattle can be attributed (add in word)

Author response 16: Corrections made

Reviewer comment 17: Line 76- Due to the lack of availability of a natural diet (add in word)

Author response 17: Corrections made

Reviewer comment 18: Line 81- not sure that sip is the correct word- maybe slip or seep

Author response 18: “Sip” has been replaced by “enter.”

Reviewer comment 19: Line 82- do you have any data on cattle performance, usage or general health based on what you have said. Is there any breed data too as this would be interesting, or any data on sex?

Author response 19: In India, people use cow/cattle for milk, meat, agriculture, cart pulling, cow dung used for manure, plow, etc. Further, the street cows are used mainly for milk. Usually, male street cattle do not have owners. In contrast, a majority of cows seen on the street have owners. However, space crunch around the houses compels the owners to release their cows on the street either permanently or temporarily. But, whenever a cow gives birth to the calves and starts lactating, the respective owners take them from the roads for milking purposes during morning and evening hours. After that, again, the owners release the cows onto the streets. The city people let the dried cows and male calves free on the roads permanently.

Further, contamination of milk of urban cattle with traces of detergents, bleaching agents, fertilizers, and cancer-causing agent ‘dioxin’ is well known.

Most of the cattle present in India are related to the indigenous species Bos indicus or the exotic species Bos taurus. The former includes breeds, namely Kosali, Sahiwal, Gir, Red Sindhi, Ongole, Tharparkar, whereas the latter includes Holstein Friesian and Jersey.

We have included the above information in the revised text at the appropriate places of the article.

We did not study the performance of street cattle.

Data on sex is not available for street/stray cattle [20th Livestock Census]. 

Reviewer comment 20: Line 84- or riding two/four-wheelers (make plural)

Author response 20: Corrections made

Reviewer comment 21: Line 86- it would be nice to see this referenced with different countries

Author response 21: We have added names of the counties in the main revised text as "countries such as India, Ethiopia, Kenya, Shri Lanka, and Nepal” and have given the references.

Reviewer comment 22: Line 89- mostly based on a door to door inventory (add in word)

Author response 22: Corrections made

Reviewer comment 23: Line 99- The Lincoln index is one of the most widely used (add in words)

Author response 23: Corrections made

Reviewer comment 24: Line 115- One can estimate the total population by multiplying the density to the entire area or distance/length of the routes – this is unclear and I cant help to reword it as it doesn't make sesne

Author response 24: We have rephrased the sentence for a better understanding.

Reviewer comment 25: Line 125- transects over a 226 km2 area of the city (add in word)

Author response 25: Corrections made

Reviewer comment 26: Line 129 -and within an adjacent 10 meters on both sides (add in word)

Author response 26: Corrections made

Reviewer comment 27: Line 134- Raipur city to find out the routes of highways/roads/streets (make plural)

Author response 27: Corrections made

Reviewer comment 28: Line 136-139- I found this difficult to understand. Please reword- In this study, we included the grids that covered ≥ 50% of the area (i.e., 163 grids: 151 grids covered 75-100% and 12 grids covered 50-75%). We excluded the remaining 27 grids (i.e., the grids those covered < 50% of area).

Author response 28: We have rephrased the sentence for a better understanding.

Reviewer comment 29: Line 153- highways/roads/streets-(make all plural)

Author response 29: Corrections made

Reviewer comment 30: Line 156- using a Nikon D-3400 still camera (add in word)

Author response 30: Corrections made

Reviewer comment 31: Line 163- was this data noted for all cattle seen?

Author response 31: Yes, we noted down the biographical (gender, age class), morphological (body color, horn shape & size, udder size), and additional characteristics (collar, bells, and cut mark in the ear) of each cattle were also noted down on the spot during sampling hours.

Reviewer comment 32: Table one- calf not calve

Author response 32: Corrections made

Reviewer comment 33: Line 216- average number of street cattle in a unit square kilometer area (add in word)

Author response 33: Corrections made

Reviewer comment 34: Line 222-224 could be combined to aid reading flow

Author response 34: We employed one way ANOVA and Mann-Whitney U test to compare the density of cattle in 20 grids, and the frequency of male and female cattle within each age group, respectively, at p ≤ 0.05. Besides, we derived the sex ratio from the number of male street cattle divided by female street cattle.

We have rephrased the above sentence.

Reviewer comment 35: Line 228- calf rather than calve

Author response 35: Changes made

Reviewer comment 36: Line 231- add in manufacturer for SPSS

Author response 36: We have added manufacturer for SPSS: “IBM SPSS Statistics for Windows, Version 20.0 (IBM Corp., Armonk, N.Y., USA)”

Reviewer comment 37: Line 262- in the city

Author response 37: Corrections made

Reviewer comment 38: Line 268-269- Cattle per km in the different studied grids was significantly different (reword)

Author response 38: Reworded

Reviewer comment 39: Line 281- calf rather than calve

Author response 39: Changes made

Reviewer comment 40: Line 292- For each female, there was less than one male in all 20 studied grids.

Author response 40: Changes made

Reviewer comment 41: Why were different values set for significant for the p values?

Author response 41: When we compare the mean values of parametric/continuous data (here, it was cattle per km) of more than two groups, we use ANOVA. While performing the ANOVA (using SPSS), we obtain an output of in-built pairwise comparisons by using post hoc tests (here we opted for Duncan's multiplerange test). Therefore, we set only a single significance level (i.e., p ≤ 0.05).

In contrast, if we compare more than two groups having nonparametric/frequency data (here, it was the number of cattle), we use nonparametric ANOVA (i.e., the Kruskal-Wallis rank test) and we set the significance level with 95% confidence level (p ≤ 0.05) to evaluate the overall difference. Further, if we want to look for pairwise comparisons, we do not have in-built post hoc tests. Therefore, we perform the Mann-Whitney U test to determine significant pairwise differences for each possible pairs. It implies that instead of rejecting one null hypothesis, we reject n number of null hypotheses for n pairs of averages. This procedure compounds the chances of committing type-1 errors.

To overcome the problem of increasing the error rate, we set a new alpha level (p-value) by Bonferroni correction. Then, we decide on the significant differences in each paired group from a new p-value. Therefore, we set different p-values to determine the exact significant differences.

Please note that in the section ‘Age composition' of the result part, we have made one necessary correction in the revised manuscript. In the original paper, we had used p ≤ 0.008 (0.05/6) while comparing the paired group. The correct significant level will be p ≤ 0.003 (0.05/15) after Bonferroni corrections. The reason is that we have divided the original alpha level (p ≤ 0.05) by the number of groups (i.e., 6). But, the actual initial alpha level should be divided by the number of possible pairs (i.e., 15) while comparing six groups.

Reviewer comment 42: Line 306- For this, initially, the significant level used for overall mean differences (reword)

Author response 42: Reworded

Reviewer comment 43: Line 308- significance level

Author response 43: Changes made

Reviewer comment 44: Line 315- The result of the Kruskal-Wallis test shows that there was significant (reword)

Author response 44: Reworded

Reviewer comment 45: Line 318- The significant level (reword)

Author response 45: Reworded

Reviewer comment 46: Line 319- The frequency of cows in 1 km2 grid (make plural)

Author response 46: Changes made

Reviewer comment 47: Line 319- you confused me a bit here by using cows compared to the other group. Can you define what you mean by a cow, and how you identified one? You have previously used cows to encompass all of the age groups

Author response 47: We used the term 'cattle' to encompass all of the age groups, such as cow, bull, heifer, immature male, female calf, and male calf. We used the word 'cow' for mature female cattle only. We identified cows by looking at their udder and teat size.

Reviewer comment 48: Line 322- The frequency of a bull was (add in word)

Author response 48: We rectified all grammatical errors. “The frequency of bulls.”

Reviewer comment 49: Line 325- Further, the frequency of heifers (add in word and make plural)

Author response 49: We rectified all grammatical errors.

Reviewer comment 50: Line 343- Based on this finding, we can interpret that for every 54 people in the city, there is at least 1 street cow. (reword)

Author response 50: Reworded

Reviewer comment 51: Line 368- line and block do not need capitalising- and throughout

Author response 51: Changes made

Reviewer comment 52: Line 402- having a lower number of residential households (3100) (reword and make plural)

Author response 52: Changes made

Reviewer comment 53: Line 405- possible reasons for a lower density of the street cattle (reword)

Author response 53: We rectified all grammatical errors.

Reviewer comment 54: Line 408- On the other hand, the Daganiya area had the maximum density of street cattle. (reword)

Author response 54: Reworded

Reviewer comment 55: Line 418- They have a high potential for breeding (reword)

Author response 55: Reworded

Reviewer comment 56: Your conclusion isn't a conclusion, more a summary. Please write an appropriate conclusion which may deal with control of the numbers etc. Is there any risk posed for disease transmission?

Author response 56: We have modified the Section ‘conclusions' in the revised manuscript.

We have revised the article, taking into consideration all inputs and suggestions of the esteemed reviewers. We do hope that the revised paper will be up to the expectations of the honorable reviewers and editors.

---

## [Decision Letter · Decision Letter 1]

7 Oct 2020

PONE-D-20-16241R1

A population estimation study reveals a staggeringly high number of cattle on the streets of urban Raipur in India

PLOS ONE

Dear Dr. Parganiha

Thank you for submitting your manuscript to PLOS ONE. After careful consideration, we feel that it has merit but does not fully meet PLOS ONE’s publication criteria as it currently stands. Therefore, we invite you to submit a revised version of the manuscript that addresses the points raised during the review process.

Many thanks for submitting your manuscript to PLOS One

Your manuscript was reviewed by the same two experts who reviewed the original submission and they have recommended a few minor changes be made prior to acceptance.

As these are generally only minor grammatical or typographical issues, please do not write a full response for these, just a line saying that they have been done will suffice, Any larger comments please write a response to.

I wish you the best of luck with your revisions

Hope you are keeping safe and well in these difficult times

Thanks

Simon

We look forward to receiving your revised manuscript.

Kind regards,

Simon Clegg, PhD

Academic Editor

PLOS ONE

Reviewers' comments:

Reviewer's Responses to Questions

**Comments to the Author**

1. If the authors have adequately addressed your comments raised in a previous round of review and you feel that this manuscript is now acceptable for publication, you may indicate that here to bypass the “Comments to the Author” section, enter your conflict of interest statement in the “Confidential to Editor” section, and submit your "Accept" recommendation.

Reviewer #1: (No Response)

Reviewer #2: All comments have been addressed

2. Is the manuscript technically sound, and do the data support the conclusions?

Reviewer #1: Partly

Reviewer #2: Yes

3. Has the statistical analysis been performed appropriately and rigorously? 

Reviewer #1: Yes

Reviewer #2: Yes

4. Have the authors made all data underlying the findings in their manuscript fully available?

Reviewer #1: Yes

Reviewer #2: Yes

5. Is the manuscript presented in an intelligible fashion and written in standard English?

Reviewer #1: No

Reviewer #2: Yes

6. Review Comments to the Author

Reviewer #1: Introduction

L.91 - 100 Text added needs to be check for grammar and to be rephrased to be more sound. Any reference is missing.

M and M

Changes have been made accordingly

L.246 ...formula in denominator maybe replace Total number by Total area (space) when measured in square km.

Results

No change

Discussion

No change

Conclusions

Conclusion part was extended but from my point of view it is more extended summary than real Conclusions.

I have to repeat that this is the weakest part of the manuscript. For innovation it is not enough to that mention ...this is the first study made....

Authors adopted some hints but ideas were not extended in Conclusions in meaning of broader usage nor made reader trust that the one of the methodologies used could be considered as "gold" standard or not.

Reviewer #2: This is an improved manuscript, which reads well. I have a few very minor comments below, with the main ones about the conclusion.

Line 55- maybe saying the largest or highest zoomass may be better?

Line 83- In contrast, a majority of the cows seen on…. (add in word)

Line 95- is Sri Lanka spelt as you have it? Sorry American English so just wanted to check

Line 416-417- a bit repetitive, consider deleting the last sentence

Line 430- the conclusion is still a bit of a summary- how does your study help to overcome human cattle interaction issues?

Why is the study important, could it help prevent interactions? Control disease? Be useful for population control etc? Things like that maybe worth mentioning.

7. PLOS authors have the option to publish the peer review history of their article (what does this mean?). If published, this will include your full peer review and any attached files.

Reviewer #1: No

Reviewer #2: No

---

## [Author Response · Author response to Decision Letter 1]

19 Nov 2020

Response to Reviewer(s)' Comments

We would like to express our gratitude towards the esteemed reviewers for providing useful suggestions. We tried our best to revise the manuscript in compliance with the valuable tips offered to us.

Reviewer #1

Reviewer comment 1: Introduction

L.91 - 100 Text added needs to be check for grammar and to be rephrased to be more sound. Any reference is missing.

Author response 1: We have rephrased the sentences to be more sound and have added the reference.

Reviewer comment 2: M and M

Changes have been made accordingly

L.246 ...formula in denominator maybe replace Total number by Total area (space) when measured in square km.

Author response 2: Corrections made

Reviewer comment 3: Conclusions

Conclusion part was extended but from my point of view it is more extended summary than real Conclusions.

I have to repeat that this is the weakest part of the manuscript. For innovation it is not enough to that mention ...this is the first study made....

Authors adopted some hints but ideas were not extended in Conclusions in meaning of broader usage nor made reader trust that the one of the methodologies used could be considered as "gold" standard or not.

Author response 3: We have further modified the section 'conclusions' in the revised manuscript.

Reviewer #2:

Reviewer comment 4: This is an improved manuscript, which reads well. I have a few very minor comments below, with the main ones about the conclusion.

Author response 4: We express our special thanks to the honorable reviewer to appreciate our revised manuscript.

Reviewer comment 5: Line 55- maybe saying the largest or highest zoomass may be better?

Author response 5: We have modified the statement as "In the animal world, cattle represent the largest zoomass."

Reviewer comment 6: Line 83- In contrast, a majority of the cows seen on…. (add in word)

Author response 6: Corrections made

Reviewer comment 7: Line 95- is Sri Lanka spelt as you have it? Sorry American English so just wanted to check

Author response 7: Corrections made

Reviewer comment 8: Line 416-417- a bit repetitive, consider deleting the last sentence

Author response 8: We have deleted the last repetitive sentence.

Reviewer comment 9: Line 430- the conclusion is still a bit of a summary- how does your study help to overcome human cattle interaction issues?

Why is the study important, could it help prevent interactions? Control disease? Be useful for population control etc? Things like that maybe worth mentioning.

Author response 9: We have further modified the section 'conclusions' in the revised manuscript.

We have revised the manuscript, taking into consideration all inputs and suggestions of the esteemed reviewers. We do hope that the revised manuscript will be up to the expectations of the honorable reviewers and editors.

---

## [Editor Report · Decision Letter 2]

25 Nov 2020

A population estimation study reveals a staggeringly high number of cattle on the streets of urban Raipur in India

PONE-D-20-16241R2

Dear Dr. Parganiha,

We’re pleased to inform you that your manuscript has been judged scientifically suitable for publication and will be formally accepted for publication once it meets all outstanding technical requirements.

Kind regards,

Simon Clegg, PhD

Academic Editor

PLOS ONE

Additional Editor Comments:

Many thanks for resubmitting your manuscript to PLOS One

As you have addressed all comments from the reviewers and the manuscript reads well, I have recommended it for publication

You should hear from the Editorial Office shortly

It was a pleasure working with you and I wish you all the best for your future research

Hope you are keeping safe and well in these difficult times

Thanks

Simon

---

## [Editor Report · Acceptance letter]

1 Dec 2020

PONE-D-20-16241R2 

A population estimation study reveals a staggeringly high number of cattle on the streets of urban Raipur in India 

Dear Dr. Parganiha:

I'm pleased to inform you that your manuscript has been deemed suitable for publication in PLOS ONE. Congratulations! Your manuscript is now with our production department. 

Kind regards, 

on behalf of

Dr. Simon Clegg 

Academic Editor

PLOS ONE